# Visible Light Spectrum Extraction from Diffraction Images by Deconvolution and the Cepstrum

**DOI:** 10.3390/jimaging7090166

**Published:** 2021-08-28

**Authors:** Mikko E. Toivonen, Topi Talvitie, Chang Rajani, Arto Klami

**Affiliations:** Department of Computer Science, University of Helsinki, 00560 Helsinki, Finland; topi.talvitie@iki.fi (T.T.); chang.rajani@helsinki.fi (C.R.); arto.klami@helsinki.fi (A.K.)

**Keywords:** spectrum, spectrometer, cepstrum, deconvolution, diffraction imaging

## Abstract

Accurate color determination in variable lighting conditions is difficult and requires special devices. We considered the task of extracting the visible light spectrum using ordinary camera sensors, to facilitate low-cost color measurements using consumer equipment. The approach uses a diffractive element attached to a standard camera and a computational algorithm for forming the light spectrum from the resulting diffraction images. We present two machine learning algorithms for this task, based on alternative processing pipelines using deconvolution and cepstrum operations, respectively. The proposed methods were trained and evaluated on diffraction images collected using three cameras and three illuminants to demonstrate the generality of the approach, measuring the quality by comparing the recovered spectra against ground truth measurements collected using a hyperspectral camera. We show that the proposed methods are able to reconstruct the spectrum, and, consequently, the color, with fairly good accuracy in all conditions, but the exact accuracy depends on the specific camera and lighting conditions. The testing procedure followed in our experiments suggests a high degree of confidence in the generalizability of our results; the method works well even for a new illuminant not seen in the development phase.

## 1. Introduction

In everyday speech, we use the word “color” to refer to the perceived appearance of objects, using named color categories such as “red” that represent a broad range of visible appearances. *Color spaces* such as RGB or CMYK can be used to define specific colors, but there is no universal mapping from perceived color to these spaces since the former depends on the lighting and the observer. Both natural (humans and animals) observers and digital sensors capture light—and hence colors—differently. The CIEL*a*b* color space, based on a standard observer, is a good approximation for a perceptually uniform color space, but for still more comprehensive characterization of color, we can consider the *spectrum* an object reflects. Together with the light spectrum and a perception model, this is enough for determining the perceived color in all conditions.

The conventional way of determining the absorption spectrum is *spectroscopy*, typically carried out by sensors that record the intensity of light for each wavelength separately to form the full intensity spectrum. For the measurement of light, a spectroradiometer is typically used, while spectrophotometers are used to measure the reflectance spectra of objects [1]. We, instead, considered the task of determining the light spectrum with imaging. Imaging-based methods have also been studied for a long time [2,3,4,5,6], but the classical solutions are relatively complex optical devices. In contrast to these works, we used ordinary digital CMOS sensors designed to mimic human perception. We acquired an image with a camera that only records the color information in the RGB color space in some lighting conditions, and the task was to infer the full reflectance spectrum of the imaged object from a single image or multiple images of the same scene, using a computational algorithm that reconstructs the spectrum that is not directly measured by the device. Compared to the classical approaches, this has the advantage of having significantly lower cost as complicated hardware is replaced with computational algorithms.

The most relevant previous work considered the task of estimating the reflectance spectrum with a computational method from an image acquired with a monochromatic or a trichromatic camera, typically equipped with passive color filters. Several methods combining multispectral imaging with algorithms that estimate the reflectance spectrum (and subsequently, the color) by explicitly assuming it to be smooth were proposed [7,8,9,10]. Similarly, the observation that most of the variance in the reflectance spectra can be explained by a small number of principal components[7,11] has been used for estimating them using principal component analysis of multispectral images [8,12]. Other works assumed that Gaussian distributions are good estimates for reflectance spectra [10,13] and used these for the recovery of the spectrum. An alternative to using color filters is to use multiple illuminants for multispectral image capture [14]. A dedicated hyperspectral camera can also be used for estimating the radiance spectrum, as well as the reflectance spectrum with an additional calibration for the illuminant spectrum [15]. While recovering the reflectance spectrum from a hyperspectral image can be easy in controlled environments, greater care is needed when imaging uncontrolled outdoor scenes [16].

Our approach shares the conceptual motivation of the previous research covered above, but uses a simpler optical construction and makes less assumptions about the spectra. In particular, we used a purely passive add-on device attached to ordinary consumer cameras and did not require knowing its spectral response, and our computational method requires only mild smoothness assumption, in contrast to the strong parametric assumptions of, e.g., [8,10,12,13]. The core idea of our approach is in the use of a specific diffractive element—a transmissive diffraction grating—placed in front of the camera sensor. The diffraction displaces light at different wavelengths spatially across the image area. By carefully designing the diffraction element, which also includes a suitably placed field stop, we can control the diffraction in a manner that allows computational reconstruction of the spectrum. By using broad-spectrum lighting (e.g., halogen illumination) and assuming a single reflectance spectrum per object, we can then provide an estimate for the full radiance spectrum of the imaged object. Diffraction images have previously been used, e.g., for the acquisition of hyperspectral images [17,18,19,20], camera sensor response calibration [21,22], multiplane imaging [23], and single-camera 3D imaging [24,25]. Similar to our method, Javoršek et al. [26] used a diffraction grating together with a biconvex lens for camera sensor spectrum response estimation. These works focused on specific imaging configurations, whereas we provide the first practical method for estimating the radiance spectrum with demonstrated high accuracy across different lighting conditions and imaging devices.

This paper introduces both the optical element that performs diffraction in a controlled manner and two alternative computational algorithms for reconstructing the spectrum. Both methods are based on supervised machine learning trained to reconstruct the spectrum from specific forms of signal representation, but differ in terms of the initial processing of the diffraction image. The first approach uses the deconvolution operation to revert the diffraction, whereas the latter relies on computing the cepstrum of the image [27].

We empirically validated the approach with experiments specifically designed to highlight its generality in terms of imaging devices and lighting conditions. Rather than demonstrating the technique using a single prototype device, we conducted experiments using three different types of cameras—a smartphone, a DSLR, and a camera module for embedded imaging applications—using three different lighting conditions. By comparing against the ground truth spectrum obtained by a dedicated hyperspectral camera, we show that the approach can estimate the spectrum under two test broad-spectrum lighting conditions and for all three cameras. The exact accuracy naturally depends on the camera and the lighting, and for some conditions, the reconstructions were less accurate especially for longer wavelengths for which the sensitivity of typical camera sensors is worse; however, the results demonstrated that the same general principle works for all cases. We used a strict testing policy where the final test set was kept secret from the persons conducting all computational experiments until the final evaluation. The main result was that the proposed approach can estimate the reflectance spectrum, and consequently the color, in the studied lighting conditions. In terms of the computational algorithms, the method based on the deconvolution operation was found to be more accurate and is to be preferred in practice.

The rest of the manuscript is organized as follows. We first introduce the foundations for diffraction imaging and the computational analysis of diffraction images in Section 2. Building on this, we proceed to introduce our concrete approach, first explaining the computational algorithms for reconstructing the spectrum in Section 3 and the specific imaging setup in Section 4, followed by additional details on the implementation of the computational approach and the testing setup in Section 5. Section 6 presents the results with the discussion, followed by the conclusions in Section 7.

## 2. Foundations

Our approach was based on a combination of an imaging device that disperses light with a diffractive element to distribute the spectral information spatially and a computational method for reconstructing the spectrum from the resulting image. We first provide a brief overview to diffraction and explain the specific form of diffraction images we used, then discuss the general principle for computational modeling of diffraction images by representing them as the results of convolution with a diffraction kernel. This provides the necessary background for understanding the specific computational methods presented in Section 3.

### 2.1. Diffraction Imaging

In our work, the phrase *diffraction imaging* refers to taking photographs with a photographic camera that has a transmissive diffraction grating placed on its optical path. Previously, diffraction images have been used for spectral and hyperspectral imaging [17,20,28,29], as part of the so-called *computed tomography spectral imaging (CTSI)* approach, as well as for estimating camera spectral responses [21]. In these uses, the purpose of the diffraction grating is to disperse light into multiple dislocations at the camera’s sensor plane in relation to the wavelength of light. Assuming perpendicularly incident rays to a regular transmissive diffraction grating (We use “regular diffraction grating” to refer to a regular pattern of a rectangular or parallelogram grid. This is in contrast to an arbitrary diffraction grating pattern, for which diffraction can be described by Fourier diffraction.), the relation between the wavelength and direction of diffraction in one dimension is described by $D\sin\theta_{n}=n\lambda$, where *D* is the spacing between the grating grooves, $\theta_{n}$ is the angular direction of the diffracted light, *n* is an integer denoting the diffraction order, and λ is the wavelength of light. For small $\theta_{n}$, we can assume the small angle approximation form of $D\theta_{n}=n\lambda$, giving us a linear relation between the direction of diffracted light and the wavelength of light. Two-dimensional diffraction gratings function in a similar way, producing diffraction components in directions defined by the axes of the diffraction grating grooves. While the extent of diffraction, i.e., the extent of spatial dislocation, is approximately defined by $D\theta_{n}=n\lambda$, the exact geometry of the diffraction grating determines the shape of the diffraction kernel h(·,·,λ). Figure 1 shows two diffraction images of point-like illuminants: one for monochromatic red light and one for broadband white light. The diffraction image in Figure 1a reveals the shape of the diffraction kernel for a single wavelength $\lambda_{m}$, while Figure 1b reveals the shape of the diffraction kernel by superimposing all the wavelengths onto a single image.

For a naive diffraction image, acquired using a diffraction grating with no additional optical elements, the zeroth-order diffraction components, i.e., for n=0, are superimposed with higher-order diffraction components, i.e., for n>0; a specific pixel recording may correspond to either zeroth-order component or a higher-order component with dislocation. Even though some methods attempt to handle this by using global [17] or local [30] image structure, we prefer using field stops, similar to [19], to eliminate the overlaps. By limiting the scene to what is visible through the field stop, we can enforce all higher-order diffraction components to be outside of the area covered by the zeroth-order components; see Figure 2 for an illustration of the images used in our setup.

### 2.2. Modeling Diffraction Images

Mathematically, the production of a diffraction image can be expressed as a convolution with the diffraction kernel of the grating element. We assumed that the image consists of only one color spectrum, which can be achieved using a black mask, i.e., a mask essentially void of color. We denote the convolution of two functions *f* and *g* by f∗g and frequently use Fourier transforms for manipulating and analyzing convolutions because of the convenient property: a Fourier transform of the convolution can be computed as the product of the transforms for the two functions. That is, the Fourier transform of (f∗g)(x,y) is $F\left(f_{x},f_{y}\right)G\left(f_{x},f_{y}\right)$, where $F(f_{x},f_{y})$ is the Fourier transform of f(x,y) and $G(f_{x},f_{y})$ of g(x,y) and $f_{x}$ and $f_{y}$ are the frequency parameters for the *x* and *y* dimensions, respectively. Denoting wavelengths by λ and spatial coordinates with variables *x* and *y*, for the diffraction image d(x,y), illuminant intensity distribution i(λ), scene reflectance $r_{\lambda }\left(x,y\right)=r\left(x,y,\lambda \right)$, camera spectral sensitivity s(λ), and the diffraction kernel $h_{\lambda }\left(x,y\right)=h\left(x,y,\lambda \right)$, we have for each color c∈{R,G,B}:(1)$$\begin{array}{cc} d_{c}\left(x,y\right) & =\int_{\lambda_{min}}^{\lambda_{max}}i\left(\lambda \right)s_{c}\left(\lambda \right)\left(h_{\lambda }*r_{\lambda }\right)\left(x,y\right)d\lambda \end{array}$$

This formulation assumes flat illumination i(λ) such that illumination intensity is spatially constant and is only a function of wavelength λ. The diffraction kernel $h_{\lambda }\left(x,y\right)$ is assumed to be linear w.r.t. to the intensity of light and isoplanatic and oriented parallel to the imaging sensor. Furthermore, the model, and subsequent models herein, assumes the image is free of noise. This does not strictly hold in real imaging conditions due to electronic and shot noise, but in practice, the overall method will later be shown to work well even if taking images without any noise cancellation functionality.

Since we are not concerned with estimating the reflectance, but instead just the spectra, because reflectance spectra can be calculated from a given reflected spectrum by knowing the spectral power distribution of the illuminant, we can reduce the product of the illuminant intensity distribution and scene reflectance to $u_{\lambda }=u\left(x,y,\lambda \right)=i\left(\lambda \right)r\left(x,y,\lambda \right)$, so that we have:(2)$$\begin{array}{cc} d_{c}\left(x,y\right) & =\int_{\lambda_{min}}^{\lambda_{max}}s_{c}\left(\lambda \right)\left(h_{\lambda }*u_{\lambda }\right)\left(x,y\right)d\lambda . \end{array}$$

For a monochromatic illuminant at wavelength $\lambda_{m}$, the diffraction image reduces to a single convolution defined by the diffraction kernel at $\lambda_{m}$ and the product of the sensor sensitivity, scene reflectance, and illuminant intensity at $\lambda_{m}$:$$\begin{array}{cc} d_{c,\lambda_{m}}\left(x,y\right) & =s_{c}\left(\lambda_{m}\right)\left(h_{\lambda_{m}}*u_{\lambda_{m}}\right)\left(x,y\right). \end{array}$$

After calibration, estimating the wavelength $\lambda_{m}$ would then be a simple matter of determining the extent to which the higher-order diffraction components have been dislocated in relation to the zeroth-order diffraction component. Unfortunately, this simple computation does not generalize directly for arbitrary illuminants. Instead, we need dedicated computational algorithms, explained next.

## 3. Method

In this section, we present two computational algorithms for reconstructing the spectrum from diffraction images acquired using the imaging setup explained later in Section 4. The first approach is based on directly reverting the convolutional model (Equation 2) for the diffraction image using the *deconvolution* operation, whereas the latter computes the *cepstrum* of the image and uses it for extracting the spectrum, building on the desirable mathematical properties of cepstra [27]. Both methods contain some elements that are trained using supervised machine learning, based on a collection of diffraction images for which ground truth spectra are available, carried out using appropriately chosen algorithms for each of the methods.

### 3.1. Algorithm 1: Deconvolution

Since we modeled the diffraction images $d_{c}\left(x,y\right)$ as convolutions of u(x,y), a natural approach for extracting the spectrum is to invert the convolution with the *deconvolution* operation. Algorithm 1 builds on this basic intuition, solving the deconvolution problem—for the specific types of diffraction images considered here—in the space of Fourier transforms.

Assume that the image consists of only one color spectrum with varying brightness; more exactly, there exists a single spectrum *f* and a per-pixel scene brightness map *b* such that the hyperspectral image u(x,y,λ) is given by the product f(λ)b(x,y). For the derivation of the algorithm, we assumed this to hold, and for practical imaging, the assumption can be satisfied by using a black mask that blocks the areas outside the color patch. By substituting the product form of *u* into (Equation 2), we obtain:$$\begin{array}{cc} d_{c}\left(x,y\right) & =\int_{\lambda_{min}}^{\lambda_{max}}s_{c}\left(\lambda \right)f\left(\lambda \right)\left(h_{\lambda }*b\right)\left(x,y\right)d\lambda . \end{array}$$

By interchanging the order of integration and convolution, we see that $d_{c}=p_{c}*b$, where:(3)$$\begin{array}{cc} p_{c}\left(x,y\right) & =\int_{\lambda_{min}}^{\lambda_{max}}s_{c}\left(\lambda \right)f\left(\lambda \right)h_{\lambda }\left(x,y\right)d\lambda . \end{array}$$

The convolution kernel $p_{c}$ is the diffraction pattern of a point with spectrum *f* generated by the whole system, taking into account the diffraction kernel *h* of the diffraction grating, and the camera spectral sensitivity $s_{c}$. Figure 1b shows an example of such a diffraction pattern.

We recovered the color spectrum *f* in two phases. In the first phase, we solved $p_{c}$ for all colors *c* from the convolution equation $d_{c}=p_{c}*b$, using deconvolution. The deconvolution was possible because we knew both $d_{c}$ and *b*: the diffraction image $d_{c}$ is directly measured by the camera and the brightness map *b* can also be extracted from the intensities of the center component of the diffraction image (provided that the diffraction components do not overlap). In the second phase, we extracted the spectrum *f* from the system diffraction pattern $p_{c}$ using the fact that for each wavelength $\lambda \in [\lambda_{min},\lambda_{max}]$, the monochromatic diffraction pattern $h_{\lambda }$ is nonzero only close to a small number of isolated peaks, one per diffraction component. Thus, by combining (Equation 3) with suitable smoothness assumptions for $s_{c}$ and *f*, we can estimate $f\left(\lambda \right)\approx S\left(x,y,\lambda \right)p_{c}\left(x,y\right)$ for each peak (x,y) of $h_{\lambda }$, where the coefficients S(x,y,λ) depend only on the camera spectral response $s_{c}$ and grating pattern $h_{\lambda }$, and hence can be calibrated per imaging system. For many trichromatic cameras, the camera spectral response $s_{c}$ for each channel can be assumed to be smooth due to the Luther condition [31]. The spectrum *f* can also be assumed smooth by assuming a smooth material reflectance spectrum [32,33,34] and a smooth spectral power distribution of the illuminant. The latter may not hold for example if using artificial LED or fluorescent illumination.

Let us now consider the first phase of solving $p_{c}$ from the convolution equation $d_{c}=p_{c}*b$. Because convolution corresponds to pointwise multiplication in the Fourier space, it holds that ${\hat{d}}_{c}={\hat{p}}_{c}\hat{b}$, where ${\hat{d}}_{c}$, ${\hat{p}}_{c}$, and $\hat{b}$ are the two-dimensional Fourier transforms of $d_{c}$, $p_{c}$, and *b*, respectively, and the arithmetic on the functions is carried out pointwise. In the absence of noise, we could simply solve ${\hat{p}}_{c}={\hat{d}}_{c}/\hat{b}$ and obtain $p_{c}$ by applying the inverse Fourier transform to ${\hat{p}}_{c}$. However, this kind of naive deconvolution does not typically handle noise in $d_{c}$ and *b* well.

We made the deconvolution more noise resistant using regularization: we obtained the estimate $p_{c}^{\prime }$ for $p_{c}$ by minimizing a cost function:$$p_{c}^{\prime }=\underset{g:\;R^{2}\to R}{\mathrm{argmin}}\left(||g*b-d_{c}{\left|\right|}_{2}^{2}+{\left||k*g|\right|}_{2}^{2}\right),$$
where $\left|\right|\cdot {\left|\right|}_{2}^{2}$ is the absolute square integral over the domain and *k* is a real-valued regularization kernel. By the Parseval theorem, we can express this for the Fourier transforms of the functions using pointwise multiplications instead of convolutions:$${\hat{p}}_{c}^{\prime }=\underset{g:\;R^{2}\to R}{\mathrm{argmin}}\left(\left|\right|\hat{g}\hat{b}-{\hat{d}}_{c}{\left|\right|}_{2}^{2}+\left|\right|\hat{k}\hat{g}{\left|\right|}_{2}^{2}\right).$$

If we allow *g* to be a complex function, then the minimization task can be carried out for each point $\xi =(\xi_{x},\xi_{y})$ in the frequency domain independently:$${\hat{p}}_{c}^{\prime }\left(\xi \right)=\underset{z\in C}{\mathrm{argmin}}\left(|z\hat{b}\left(\xi \right)-{\hat{d}}_{c}{\left(\xi \right)|}^{2}+{\left|\hat{k}\left(\xi \right)z\right|}^{2}\right).$$

Each minimizing value ${\hat{p}}_{c}^{\prime }\left(\xi \right)\in C$ can be found by solving for the value at which the partial derivatives of the cost function with respect to the real and imaginary part of *z* are zero. This calculation yields the result:(4)$${\hat{p}}_{c}^{\prime }\left(\xi \right)=\frac{\bar{\hat{b}\left(\xi \right)}{\hat{d}}_{c}\left(\xi \right)}{|\hat{b}{\left(\xi \right)|}^{2}+{\left|\hat{k}\left(\xi \right)\right|}^{2}}.$$

The estimate $p_{c}^{\prime }$ obtained from this ${\hat{p}}_{c}^{\prime }\left(\xi \right)$ using the inverse Fourier transform is actually a real function and hence a solution to the original minimization problem. This is because *b*, $d_{c}$, and *k* are real functions, which means that the reality conditions $\hat{b}\left(\xi \right)=\bar{\hat{b}\left(-\xi \right)}$, ${\hat{d}}_{c}\left(\xi \right)=\bar{{\hat{d}}_{c}\left(-\xi \right)}$, and $\hat{k}\left(\xi \right)=\bar{\hat{k}\left(-\xi \right)}$ hold, and they imply the reality condition for ${\hat{p}}_{c}^{\prime }$ as well.

Figure 3 shows an example of a diffraction pattern *p* obtained using 2D deconvolution (Equation 4) from a diffraction image. While the diffraction pattern is clearly visible, the image contains a significant noise pattern outside the diffraction components, which may skew the signal in the actual diffraction patterns. To avoid this, we chose to limit the degrees of freedom in the deconvolution by using a 1D deconvolution instead.

From the diffraction pattern obtained using 2D deconvolution, it is easy to detect the diffraction directions by finding the angles in which the total brightness is highest. Let us consider the modified instance of the problem where we detected the diffraction directions, rotated the diffraction image such that one of the first-order diffractions is made horizontal, and cropped out all the other diffraction components from the image. Now, the problem is 1D in the sense that the diffraction pattern $p_{c}\left(x,y\right)$ should be zero if y≠0.

Let us now reformulate the deconvolution optimization problem in terms of 1D convolution. Consider the diffraction pattern $p_{c}$ to be a one-dimensional function, replacing $p_{c}\left(x,0\right)$ by $p_{c}\left(x\right)$. Denote $b_{y}\left(x\right)=b\left(x,y\right)$ and $d_{c,y}\left(x\right)=d_{c}\left(x,y\right)$. Now, the estimate $p_{c}^{\prime }$ for $p_{c}$ is obtained by minimizing a cost function as follows:(5)$$p_{c}^{\prime }=\underset{g:\;R\to R}{\mathrm{argmin}}\left(\int_{-\infty }^{\infty }||g*b_{y}-d_{c,y}{\left|\right|}_{2}^{2}\;dy+{\left||k*g|\right|}_{2}^{2}\right).$$

If we allow *g* to be a complex function and rewrite the convolutions using pointwise multiplications of 1D Fourier transforms, we obtain that:$${\hat{p}}_{c}^{\prime }\left(\xi \right)=\underset{z\in C}{\mathrm{argmin}}\left(\int_{-\infty }^{\infty }|z{\hat{b}}_{y}\left(\xi \right)-{\hat{d}}_{c,y}{\left(\xi \right)|}_{2}^{2}\;dy+{\left|\hat{k}\left(\xi \right)z\right|}_{2}^{2}\right).$$

By solving for the point at which the partial derivatives are zero, we obtain the optimum:(6)$${\hat{p}}_{c}^{\prime }\left(\xi \right)=\frac{\int_{-\infty }^{\infty }\bar{{\hat{b}}_{y}\left(\xi \right)}{\hat{d}}_{c,y}\left(\xi \right)\;dy}{\int_{-\infty }^{\infty }|{\hat{b}}_{y}{\left(\xi \right)|}^{2}\;dy+{\left|\hat{k}\left(\xi \right)\right|}^{2}}.$$

Similarly to (Equation 4), the estimate $p_{c}^{\prime }$ obtained from this ${\hat{p}}_{c}^{\prime }$ using the inverse Fourier transform is actually a real-valued function, and thus, it is also a solution to the original minimization problem (Equation 5).

### 3.2. Algorithm 2: Cepstrum

Instead of solving the deconvolution, we can alternatively approach the problem by using the cepstrum operation that has been used, e.g., in pitch determination [35], signal detection and extraction [36], image registration [37,38], and echo detection and removal [39]. The cepstrum operator has interesting properties that make it convenient for relative spectra extraction. Following [27], we summarize the most notable properties as:Scaling, i.e., multiplication, of the diffraction image affects only the center value of the convolution image;Biasing, i.e., addition, the diffraction image leads to a biasing in the convolution image that is nonlinearly relational to the biasing factor;Convolution in the diffraction image can be expressed as the sum of two cepstrum images;Rotation of the diffraction image results in a rotation of the cepstral image;The cepstrum image is invariant to shifts in the diffraction image.

Adjusting the intensity of the illuminating light or the exposure time of the diffraction image is essentially a scaling operation, which does not, in theory, affect the cepstrum image other than at the center of the cepstrum image. The core phenomenon of diffraction can be modeled accurately using the convolution operation, which in the cepstrum image can be expressed as the sum of two cepstrum images. The rotation property helps in finding the rotation of the diffraction grating in relation to the camera sensor’s axes directly from the computed cepstrum image. The cepstrum image’s invariance to shifts in the diffraction image make it easy to automatically locate and extract the salient parts of information from the cepstrum image. An example of a cepstrum image formed from the mean of 60 individual cepstrum images can be seen in Figure 1c, where the diffraction grating’s axes are at an angle with respect to the camera sensor’s axes.

In defining the cepstrum operation, we first denote $F_{2D}$ and $F_{2D}^{-1}$ as the two-dimensional Fourier and inverse Fourier transformations, respectively. The two-dimensional cepstrum $C_{2D}$, also referred to as the cepstral image, of a two-parameter function g(x,y), and denoting $G\left(f_{x},f_{y}\right)=F_{2D}\left(g(x,y)\right)$, is then defined as:$$\begin{array}{cc} C_{2D}\left(g\right)\left(x,y\right) & =F_{2D}^{-1}\left(log\left(|F_{2D}\left(g(x,y)\right)|\right)\right)\\ C_{2D}\left(g\right)\left(x,y\right) & =F_{2D}^{-1}(log\left(|G(f_{x},f_{y})|\right)\\ C_{2D}\left(g\right)\left(x,y\right) & =G_{C}\left(x,y\right). \end{array}$$

By first denoting the product of the spectral sensitivity s(λ), scene reflectance $r_{\lambda }\left(x,y\right)=r\left(x,y,\lambda \right)$, and illuminant intensity i(λ) as $q_{\lambda }\left(x,y\right)=q\left(x,y,\lambda \right)=s\left(\lambda \right)r\left(x,y,\lambda \right)i\left(\lambda \right)$, the two-dimensional cepstrum $C_{2D}$ of the diffraction image is:$$\begin{array}{cc} C_{2D}\left(d_{c}\right)\left(x,y\right) & =F_{2D}^{-1}\left(log\left(|F_{2D}\left(\int_{\lambda_{min}}^{\lambda_{max}}h_{\lambda }\left(x,y\right)*q_{\lambda }\left(x,y\right)\right)|\right)\right). \end{array}$$

Since the Fourier transform of a convolution is the product of the Fourier transforms, this becomes:$$\begin{array}{cc} C_{2D}\left(d_{c}\right)\left(x,y\right) & =F_{2D}^{-1}\left(log\left(|\int_{\lambda_{min}}^{\lambda_{max}}H_{\lambda }\left(f_{x},f_{y}\right)Q_{\lambda }\left(f_{x},f_{y}\right)|\right)\right), \end{array}$$
where $H_{\lambda }\left(f_{x},f_{y}\right)$ and $Q_{\lambda }\left(f_{x},f_{y}\right)$ are 2D Fourier transformations of $h_{\lambda }\left(x,y\right)$ and $q_{\lambda }\left(x,y\right)$, respectively. In the case of a single monochromatic light source $\lambda =\lambda_{m}$, the cepstrum image becomes:$$\begin{array}{cc} C_{2D}\left(d_{c}\right)\left(x,y\right) & =F_{2D}^{-1}\left(log\left(|H_{\lambda_{m}}\left(f_{x},f_{y}\right)Q_{\lambda_{m}}\left(f_{x},f_{y}\right)|\right)\right)\\ C_{2D}\left(d_{c}\right)\left(x,y\right) & =F_{2D}^{-1}\left(log\left(|H_{\lambda_{m}}\left(f_{x},f_{y}\right)|\right)\right)+F_{2D}^{-1}\left(log\left(|Q_{\lambda_{m}}\left(f_{x},f_{y}\right)|\right)\right)\\ C_{2D}\left(d_{c}\right)\left(x,y\right) & =C_{2D}\left(h_{\lambda_{m}}\left(x,y\right)\right)+C_{2D}\left(q_{\lambda_{m}}\left(x,y\right)\right). \end{array}$$

Thus, for a known diffraction kernel $h_{\lambda }\left(x,y\right)$, solving for $C_{2D}\left(q_{\lambda_{m}}\right)\left(x,y\right)$ is possible by computing the 2D cepstrum of the diffraction image. In practice, the computation is performed over discrete representations of diffraction images. To find the shape of the diffraction kernel, i.e., the locations where $h_{\lambda }\left(x,y\right)$ is mostly nonzero, we can use the average of the cepstra of a collection of diffraction images, similar to [17]. For regular diffraction gratings, the diffraction kernel is relatively simple and can be assumed to be known a priori with the additional assumption that the camera is not sensitive to wavelengths below and above $\lambda_{min}$ and $\lambda_{max}$.

For monochromatic light, we have an analytic relationship between the values of the cepstrum image and $q_{\lambda }\left(x,y\right)$, so that the relationship between the cepstrum image and $q_{\lambda_{m}}\left(x,y\right)$ is linear, as demonstrated in [40]. By simulation, we found that for the case of two monochromatic lights, the relation is very nearly linear if the two wavelengths are not too close to each other. For the general case, i.e., a distribution of $q_{\lambda }\left(x,y\right)$ over λ, we found that the linearity assumption between the cepstrum image and $q_{\lambda }\left(x,y\right)$ does not hold true. While estimating a higher-order analytic estimate is a difficult ordeal [41], the cepstrum image has interesting properties noted before that would make the estimation of $q_{\lambda }\left(x,y\right)$ convenient. Despite these convenient properties, there are two drawbacks: the lack of an analytical relationship between the cepstrum image and $q_{\lambda }\left(x,y\right)$ and the noisy nature of the cepstrum images.

The noise can be reduced by computing the mean of multiple cepstrum images, by capturing multiple photographs. We did not investigate the factors effecting noise thoroughly, but did study the relationship between the noise and the number of cepstrum images being averaged. One-hundred forty images were captured using the setup involving a Canon M3, described later in detail in Section 4, and the mean of all of them was used as a proxy for a noiseless image. The number of cepstrum images to be averaged was tested over the range from 1 to 80, and for each test, 100 iterations were made by randomly sampling a combination of images from the set of 140 images. The cepstrum profiles were also extracted from the proxy ground truth and the averaged cepstrum image, as described in Section 3.2.1.

Figure 4 shows the errors on a semilog scale, displaying an initially rapidly decreasing rate from 1 to approximately 20 images. The error rates continued to decline with the diminishing rate in the marginal error. Based on this result, we captured a minimum of 10 images for each color sample for the computation of the cepstrum profiles.

#### 3.2.1. Cepstrum Profile

As shown in an example cepstrum image in Figure 1c, most of the cepstrum image is dark background with higher intensity values at the axes of the image and along the cepstrum profile lines that follow the axes of the diffraction grating. If the diffraction grating’s axes are perfectly aligned with the camera sensor’s axes, the cepstrum profiles would be superimposed on the cepstrum image axes with the high intensity values at these axes dominating over the cepstrum profile. Based on this observation, the diffraction grating for each camera was rotated slightly with respect to the camera sensor’s axes so that the cepstrum profile lines would not align with the cepstrum images axes. The center of the cepstrum image contains information mostly about the center of the diffraction image and is therefore filtered out, i.e., a square area corresponding to the size of the center image in the diffraction image is set to zero in the cepstrum image. To extract the cepstrum profile, starting from the center to the edge of the cepstrum image, the profile_line function from the Python package scikit-image [42] was used, with a line width of 5 and reducing by sum. The angle of rotation of the diffraction grating was estimated using Bayesian optimization [43] by maximizing the sum of the profile line and searching from an angle range provided by manual inspection. A total of 8 cepstrum profiles corresponding to the first order diffraction components are visible in Figure 1c. However, the diagonal diffraction components had significantly lower intensity and were not used: only the cepstrum profiles for the four major direction diffraction components were extracted and used throughout.

## 4. Imaging

In this section, we describe the imaging setups for the different cameras used to obtain the diffraction images. Three cameras were used to image the same set of color samples: a Canon M3 digital single-lens reflex (DSLR) equipped with a Sigma 35mm F1.4 DG Art series lens (adapted using a EOS to EOS-M adapter), a Raspberry Pi HQ camera (Sony IMX477 sensor) with a 4mm CS mount lens, and a Samsung S10 smartphone’s main camera at f/2.4 aperture. For the collection of the training set, we used 15 discs with 18 color samples in each disc, for a total of 270 color samples. For the testing set, we used 2 discs with 18 color samples, for a total of 36 color samples. The discs were custom made, and the color samples were paint samples from three different paint manufacturers such that samples from two manufacturers (Tikkurila and Värisilmä) were used for the training set and one manufacturer (Teknos) was used for the test set. Each of the color samples were imaged using different illuminants in a darkened room. For the training set, three illuminants were used: a full-spectrum LED, halogen lamps, and a white LED, whose spectra are shown in Figure 5a. For the test set, two illuminants were used: halogen lamps and cold white LED, whose spectra are shown in Figure 5b. The test set thus included images captured using an illuminant outside of the training set to test the different models’ generalizability for illuminants not in the training set. Furthermore, all of the actual samples were different.

### 4.1. Diffraction Image

To capture diffraction images using the three cameras, each of the cameras was equipped with 2D transmissive regular diffraction grating and a field stop. Each camera was equipped with the same diffraction grating with a grating constant of approximately 191 grooves per millimeter. The diffraction gratings were placed on the optical path of the camera–lens system before the field stop. For each of the cameras, custom field stop assemblies were constructed. For the Canon M3, seen in Figure 6, a tube of approximately 12 cm in length was mounted in front of the lens with a square aperture with each side measuring 8mm at the other end of the tube. The Raspberry Pi HQ camera was placed in a custom-made rectangular box measuring 4.5 cm by 5.2 cm by 10 cm, with a square aperture each side measuring 1cm at the end opposite to the camera. For the Samsung S10 a custom-made tube 3.5 cm in length was attached to a 17 mm screw thread onto a camera case attached to the phone. The field stop in the tube for the Samsung S10 was also square with each side measuring 6 mm. For all of the cameras, we attempted to center the square field stop at the central optical path.

The masking on the color discs, as seen in Figure 6, functioned effectively as a secondary field stop at the plane where the color samples were located. The masking was somewhat poor at absorbing incident light, so a significant amount of light was still scattered and reflected from the surface of the masking, producing a low contrast for paint samples with relatively low total reflectance. However, the masking solved the problem in which different diffraction components effectively see different parts of the scene when the field stop is placed between the camera and imaged plane i.e. the imaginary plane from which the image is formed at the camera sensor. Another diffraction imaging construction was presented in [20], where a collimating lens was used to overcome the problem of scene inconsistency between diffracted orders, which for the purpose of imaging plane paint samples would introduce unnecessary optical components to the construction.

### 4.2. Imaging Setup

The imaging setup for diffraction photos consisted of the camera setup as described in Section 4.1 for each camera, a tripod mount, illuminants, color sample discs, a rotating platform, and a remote trigger mechanism. Figure 6 shows the imaging setup with the Canon M3 camera, the full-spectrum LED illuminant, and one of the color sample discs. The cameras were mounted on a tripod such that each image was captured from the same location for each camera. The cameras were placed directly over the color samples at appropriate heights so that color samples were entirely visible in the captured images. Illuminants were placed to one side in relation to the color discs and at an angle to avoid possible specular reflections from the color samples. The color sample discs were placed on a rotating platform that was accurately rotated using a stepper motor via a timing belt. The Canon M3 was remote controlled using the *gphoto2* [44] software. The Raspberry Pi HQ camera was attached to a Raspberry Pi 4 computer using the CSI interface and controlled using custom software. The Samsung S10 was remote controlled via Bluetooth using custom software on an ESP32 SOC device. In all cases, the purpose of the remote control was to trigger the photo capture without physical contact with the cameras and for the automation of image capture. The rotating platform also enabled the automation of image capturing with manual labor required only for changing the color sample discs, illuminants, and cameras.

For the Canon M3, an aperture size of f/10 was used for all images. The Raspberry Pi HQ camera was equipped with a nonadjustable aperture lens. For the Samsung S10, we used the main camera with f/2.4, based on the principle of favoring smaller apertures. All images were captured in RAW Bayer format, to maintain a linear dependence between the recorded sensor values and the intensity of light, with manual settings and manual focus. The exposure times were manually maximized such that for the first color sample, which was white, it did not produce overexposed images. All possible noise reduction and image enhancement options were switched off for image capture and recording.

For the purpose of extracting ground truth spectra, the Specim IQ hyperspectral camera was used to capture hyperspectral images of the color sample discs. The color sample discs were placed into a custom jig, and the Specim IQ camera was mounted stationary on a tripod. The illuminants were placed approximately in the same geometry, i.e., the same angle w.r.t. to the color samples. Each color sample disc and illuminant combination was imaged using the Specim IQ hyperspectral camera producing a total of 45 hyperspectral images for the training set and 4 for the testing set. The ground truth spectra were obtained from the hyperspectral images by computing the mean value in 5 × 5 neighborhoods, separately for each channel. That is, for each color patch, we computed: $$y\left(\Lambda \right)=\frac{\sum_{(x,y)\in W}HSI\left(x,y,\Lambda \right)}{25}$$
where HSI(x,y,Λ) is the value of the hyperspectral image at image coordinates (x,y) and spectral channel Λ and *W* is a 5 × 5 pixel subregion.

### 4.3. Data

The data for training and testing were collected at separate times. The data consisted of diffraction images and ground truth hyperspectral images acquired with the Specim IQ hyperspectral camera. For the training dataset, 10 diffraction images for each combination of the 3 cameras, the 3 illuminants (whose spectra are shown in Figure 5a), and 270 color patches were collected. For the deconvolution methods, the entire diffraction image was used. For the purpose of the cepstral methods, a cepstrum profile tensor was computed for each color patch as an average cepstrum profile for each of the 4 major cepstrum directions using all of the diffraction images for a color patch. The cepstrum profiles were computed from raw Bayer images without any demosaicing, but instead stacking appropriate values to their own color channels, reducing the image resolution by a factor of four. The cepstrum profile tensor for each color patch was of size $\left(\mathrm{number}\;\mathrm{of}\;\mathrm{directions},\mathrm{color}\;\mathrm{channels},n_{\lambda }\right)=\left(4,3,301\right)$, where the number of wavelength channels $n_{\lambda }$ was linearly interpolated to the inclusive range $\lambda_{min}=400\;\mathrm{nm}$ and $\lambda_{max}=700\;\mathrm{nm}$ at a 1 nm sampling interval. The cepstrum profile range was calibrated to coincide with the correct wavelength range by following the calibration procedure described in Section 5.1. Similarly, the ground truth spectrum data for each color patch were linearly interpolated to the same inclusive range, resulting in a spectrum vector of size 301. We were only concerned with relative spectra, so each ground truth spectrum y was normalized using:$$y_{\mathrm{normalized}}=yw(y),wherew\left(y\right)=\frac{\sum_{\lambda =400nm}^{700nm}y\left(\lambda \right)c\left(\lambda \right)}{\sum_{\lambda =400nm}^{700nm}c\left(\lambda \right)},and\;y\left(\lambda \right)=y_{\lambda -\lambda_{min}},$$
where c(λ) is the mean of the absolute CIE RGB color matching function [45] values at wavelength λ, as shown in Figure 7b. The purpose of the normalization was to weigh the areas of the visible spectrum that were more significant to color perception as defined by the CIE RGB color matching functions. For all the following calculations, the ground truth spectra y refer to the normalized ground truth spectra $y_{\mathrm{normalized}}$.

The test dataset was collected at a later time, rather than simultaneously with the training set, in order to simulate realistic use cases. The cameras, lights, and color patches were reset with a reasonable attempt to place them in a similar setting to how the training data were collected. In practice, the test data collection setup still differed from the training data setup, for example with differences in illuminant angles and imaging distances from the color patches. Furthermore, the color patches were selected from a different manufacturer to those in the training set, and one of the illuminants, the cold white LED, had a distinctly different spectrum from the spectra of the illuminants in the training set. Again, 10 diffraction images for each combination of the 3 cameras, the 2 illuminants (whose spectra are shown in Figure 5b), and 36 color patches were collected. Cepstrum profile tensors and ground truth spectrum data were computed in a manner identical to the training data.

The reader may be interested in the fact that the spectra of the collected color patches were smooth. Each spectrum was sampled over 102 wavelengths over the visible wavelength range of 400–700 nm by the Specim IQ hyperspectral camera used to collect the ground truth data. The full-width at half-maximum (FWHM) bandwidth, a measure of the spectral resolution, of the Specim IQ hyperspectral camera was 7 nm [46]. Furthermore, the reflectance spectra of many materials, including paints, are smooth [34], leading to the discovery that most visible reflectance spectra can be encoded by between 13 and 23 principle component parameters. Parkkinen et al. [11] found that only 8 principle components are enough to explain 99.9% of the variance in reflectance spectra. Note that this only indicates the dimensionality of the data and does not directly provide information on its smoothness, since the components may be nonsmooth. To quantify the smoothness of the spectra, we calculated the mean of the sum of absolute first and second differences of the normalized spectra, i.e., the means of $\sum_{\lambda =401nm}^{\lambda =700nm}\left|{\hat{s}}_{\lambda }-{\hat{s}}_{\lambda -1}\right|$ and $\sum_{\lambda =402nm}^{\lambda =700nm}\left|{\hat{s}}_{\lambda }-2{\hat{s}}_{\lambda -1}+{\hat{s}}_{\lambda -2}\right|$, where $\hat{s}=\frac{s}{max(s)}$ and *s* are spectra, which for the test dataset were 1.84 and 0.149, respectively. The maximum possible sum of the absolute first-order differences for a single spectrum was 150 for a comb-like spectrum, so the spectra were indeed relatively smooth. While the ground truth radiance spectra were observed to be smooth, our computational methods did not exploit this property, other than in the form of regularization, for better generalizability of recovered spectra. This was in contrast to many previous methods that explicitly assumed smooth spectra [7,8].

## 5. Computational Pipeline

This section provides the details of the full computational pipeline used for reconstructing the spectrum with the algorithms described in Section 3 from images acquired using the protocol described in Section 4. We first explain a common calibration scheme and then discuss additional elements required for applying the general algorithms for our imaging setup. Finally, we explain how the models were trained using the ground truth spectra and how the accuracy of the reconstructed spectra was evaluated.

### 5.1. Calibration

Calibration is necessary to express spectra as a function of wavelength λ. The images consisted of pixels and spectra produced by the deconvolution and cepstral method as a function of pixel dislocation $d_{p}$, which is the distance from the source of diffraction to the diffracted component in pixels. For a monochromatic point-like source, $d_{p}$ is easily measured between the zeroth-order and first-order diffraction components. We assumed that there exists a linear mapping between pixel dislocations and wavelengths. The strategy we employed to solve this linear mapping was to capture diffraction images of an illuminant through a diffuser with a known spectrum and that had multiple peaks. For the illuminant, we used an RGB LED light producing light that was in appearance white, but consisted of red, green, and blue LED light sources. Capturing one diffraction image of the RGB LED is equivalent to imaging the red, green, and blue LED separately. Other good choices for the calibration illuminant would be light sources that have multiple distinct peaks, such as some metal halide and fluorescent lamps. The spectrum of the RGB LED illuminant was measured using a Hopoocolor OHSP-350C spectral illuminance meter. From the spectrum, maxima were extracted for the red, green, and blue parts of the spectrum. As a result, we had $\left(\lambda_{i},p_{i}\right)$ pairs for i∈{red,green,blue}, and we solved for λ=ap+c, where *a* and *c* are parameters to be solved for each camera.

### 5.2. Deconvolution

To process the diffraction images for the deconvolution approach described in Section 3.1, we first needed to split each image into diffraction components. We did this using a semi-automatic method: First, we found the center image by finding the brightest point in a blurred version of the image and using a bounding area of predefined size and shape around it. Then, we obtained an estimate for the diffraction pattern using 2D deconvolution (Equation 4); this resulted in a noisy estimate similar to Figure 3. By finding the angles with the highest total brightness values in the estimated diffraction pattern, we determined the diffraction directions of the grating. By combining this with the previously calibrated pixel dislocation information, we found the clipping bounds for the diffraction components. After this, we chose one pair of diametrically opposite first-order diffraction components, rotated the image to make the diffraction horizontal, and used 1D deconvolution (Equation 6) in the rotated image to obtain an estimated 1D diffraction pattern $p_{c}^{\prime }$ for each color channel *c*. Finally, we mapped the pixel dislocations to wavelengths and averaged the two sides by setting $r_{c}^{\prime }\left(\lambda \right)=\left(p_{c}^{\prime }\left(x_{\lambda }\right)+p_{c}^{\prime }\left(-x_{\lambda }\right)\right)/2$, where $x_{\lambda }$ is the pixel dislocation corresponding to wavelength λ.

Each resulting function $r_{c}^{\prime }:\left[\lambda_{min},\lambda_{max}\right]\to R$ estimates the *raw spectrum*$r_{c}$, which is the light spectrum *f* multiplied pointwise by a response function that depends only on the spectral response $s_{c}$ of the camera and the diffraction kernel *h*, and thus staying constant in all images taken with the same imaging setup. Thus, it makes sense to train wavelength-specific linear models to obtain the estimate $f^{\prime }$ for the spectrum *f* from the raw spectrum estimates $r_{c}^{\prime }$. This means that for each imaging setup, we trained the weights $w_{c}\left(\lambda \right)\in R$ for all color channels *c* and wavelengths $\lambda \in [\lambda_{min},\lambda_{max}]$ that yielded the estimate:$$f^{\prime }\left(\lambda \right)=\sum_{c\in \{\mathrm{red},\mathrm{green},\mathrm{blue}\}}w_{c}\left(\lambda \right)r_{c}^{\prime }\left(\lambda \right).$$

For training the weights, we computed the raw spectrum estimates $r_{c}^{\prime }$ for each training image and optimized the weights to minimize the sum of squared differences between the estimates $f^{\prime }$ and the ground truth spectra *f* by solving a separate linear least squares problem for each wavelength λ. We also trained an alternative model where we constrained the weights to be non-negative using constrained linear least squares optimization.

Both deconvolutions (2D and 1D) were carried out using discretized images and discrete Fourier transforms, resulting in discretized raw spectrum estimates $r_{c}^{\prime }$ and spectrum estimate $f^{\prime }$. For the preliminary 2D deconvolution, we found that using the squared L2 norm as the regularization term, that is a one-hot regularization kernel *k*, worked sufficiently well for the purpose of detecting diffraction directions. For the final 1D deconvolution, we wanted to enforce continuity in the raw spectrum estimates, which is why we used the sum of squared differences of adjacent elements in $r_{c}^{\prime }$ as the regularization term. This means that the regularization kernel *k* was zero except for two adjacent elements *R* and −R, where *R* is a regularization weight hyperparameter, which controls the smoothness of the raw spectra obtained from the 1D deconvolution.

### 5.3. Cepstrum

As discussed in Section 3.2, to model the relationship between the cepstra and the spectra, we assumed a linear relationship. Two different models were tested: a simpler one that assumed a one-to-one mapping between elements and a more complex one where each element in the input cepstrum affected each output element. The first one is given simply by:$$\hat{y}=w\circ x+b,$$
where ∘ is an elementwise product and *w* and *b* are *l*-dimensional vectors. The other one is given by:$$\hat{y}=Wx+b,$$
where *W* is an l×l matrix and *b* is a vector of size *l*, i.e., standard linear regression.

### 5.4. Training

To measure accuracy, we used the Canberra distance, $d_{\mathrm{Can}}\left(\hat{y},y\right)=\sum_{\lambda }\frac{|{\hat{y}}_{\lambda }-y_{\lambda }|}{{\hat{y}}_{\lambda }+y_{\lambda }}$ between the spectrum extracted from diffraction images $\hat{y}$ and the ground truth spectrum y. The Canberra distance is chosen based on a comprehensive evaluation of spectral differences [47]. However, the Canberra distance does not factor in the spectral sensitivity of a camera’s sensor: a sensor’s ability to record a response for a particular wavelength range is proportional to the sensor’s spectral sensitivity. That is, if a sensor has a low sensitivity to a particular wavelength range, commonly below 400 nm or above 700 nm, one cannot expect any kind of accurate result for that range. Such reasoning also applies to the illumination spectrum as well: if there is no light at a particular wavelength range, no camera sensor response can be expected. For the aforementioned reasons, we modified the Canberra distance with a wavelength-specific weighting factor $c_{\lambda }$ and define a weighted Canberra distances as:$$\begin{array}{c} d_{\mathrm{WCan}}\left(\hat{y},y\right)=\sum_{\lambda }c_{\lambda }\frac{|{\hat{y}}_{\lambda }-y_{\lambda }|}{{\hat{y}}_{\lambda }+y_{\lambda }}. \end{array}$$

The weighting factor $c_{\lambda }$ takes into consideration the camera spectral sensitivity, the sensitivity weight, and the spectrum distribution of the illuminant, the illuminant weight, used to illuminate the color patches. Instead of measuring each camera’s spectral sensitivity, we used the mean of absolute CIE RGB color matching function [45] values, as show in Figure 7a. Absolute values were used, because the CIE color matching function for red also has negative values. The reason to use the CIE RGB color matching function values as a proxy for camera sensor sensitivity is that camera sensor spectral responses are designed to be close to these color matching functions. Indeed, a visual comparison of measured camera sensor spectral sensitivity in [48,49] to CIE RGB color matching functions reveals a close match. Any previous measurement for the camera sensor spectral sensitivity would not apply, due to the diffraction grating placed in the optical path. The final weighting factor $c_{\lambda }$ is the product of the illuminant weight and the sensitivity weight normalized to a maximum of one. For the test set illuminants, the weighting factors $c_{\lambda }$ are shown in Figure 7b, which are the product of the sensitivity weight in Figure 7a and illuminant relative spectra shown in Figure 5b. To eliminate the effect of the total power of the weighting factor, i.e., the area under the weighting factor curve, the final weighted Canberra distance is divided by the sum of $c_{\lambda }$, so the measurement for accuracy that we used becomes:(7)$$d_{WCan}\left(\hat{y},y\right)=\sum_{\lambda }c_{\lambda }\frac{|{\hat{y}}_{\lambda }-y_{\lambda }|}{{\hat{y}}_{\lambda }+y_{\lambda }}\sum_{\lambda }c_{\lambda },$$

For the deconvolution method, we needed to train two things for each camera: the weights $w_{c}\left(\lambda \right)$ for the linear transformation from the raw spectrum estimates $r_{c}^{\prime }$ to the spectrum estimate $f^{\prime }$ and the regularization weight *R*. In our case, the training data contained each ground truth spectrum *f* only in a normalized form, and thus, we also normalized our estimate spectrum $f^{\prime }$ as a postprocessing step. Due to the nonlinearity caused by the normalization, we needed to slightly alter the method described in Section 5.2 for training the weights $w_{c}\left(\lambda \right)$ for fixed *R*: we also trained a scaling coefficient for each estimate spectrum $f^{\prime }$ and alternated between minimizing the loss with respect to the weights $w_{c}\left(\lambda \right)$ and minimizing the same loss with respect to the scaling coefficients. Both of these minimization problems can be solved individually using (constrained) linear least squares, and we repeated these minimization steps until the result converged. After this, the scaling coefficients may be discarded as the spectrum estimate $f^{\prime }$ is normalized anyway.

We picked the value for the regularization weight hyperparameter *R* by optimizing the mean weighted Canberra distance in a 5-fold cross-validation. In our preliminary tests, the cross-validation score appeared to be approximately unimodal as a function of the weight *R*, and thus, we chose ternary search as the optimization algorithm.

In training for the cepstrum-based methods, the expression for the weighted Canberra distance (Equation (Equation 7)) contains absolute values and is therefore tricky to optimize. Instead, to model the relationship between the cepstra and the spectra, we optimized a simpler objective:(8)$$\begin{array}{c} d_{\mathrm{WMSE}}\left(\hat{y},y\right)=\frac{\sum_{\lambda }c_{\lambda }{\left(\hat{y}-y\right)}^{2}}{\sum_{\lambda }c_{\lambda }} \end{array}$$
which is the weighted mean-squared error. Both models presented in Section 5.3 were trained by minimizing Equation (Equation 8) using full-batch gradient descent with an adaptive learning rate. Note that the predicted spectra $\hat{y}$ need to sum to one after weighting. This was accomplished by simply dividing them by their sum at each iteration of the gradient descent. We trained each camera separately, but mixed all light sources together per camera. This allowed the model to generalize to new light sources. The results for the two presented cepstrum-to-spectrum models are given in the next section.

### 5.5. Testing Procedure

To evaluate a generally applicable method, we needed a testing procedure that evaluates the accuracy in unseen conditions. As described in Section 4.3, the test dataset consisted of color patches from a different manufacturer than the color patches in the training set. The motivation to use another paint manufacturer for the test set was to minimize the chance that the reflectance spectra of the color patches would be identical to the color patches in the training set. Two illuminants were used in the test set: halogen and cold white LED. The halogen illuminant served to test the generalizability over unseen color patches, whereas the cold white LED served to test the generalizability over both illumination conditions and unseen color patches.

The testing procedure involved three agents: a tester and two modelers (one for the deconvolution method and one for the cepstrum method). The tester did not directly partake in the modeling work, but was responsible for calculating the accuracy results over the test dataset and keeping the ground truth values secret from the modelers. During training, the modelers had access to the diffraction images, cepstrum profile tensors, and ground truth spectra for the training data, which were further split into the data used for training and validating the models. For testing, the modelers received only the diffraction images and the cepstrum profile tensors. They also knew which samples related to which camera and which illuminant, but the illuminants were identified only by Codes A and B and the illuminant spectra were not provided for them. After having selected the best models, the modelers applied them on the test images and submitted the predicted spectra for the tester. Only one additional submission was necessary for the cepstrum method due to a scaling issue, but even in this instance, the tester only revealed information for correcting the scaling.

The main goal of this strict testing protocol was to prevent even accidental fine-tuning of the models for the test conditions. The results reported in Table 1 and Table 2 were based on the submitted inferred spectra, and the models were not modified or retrained in any way after seeing the final results.

## 6. Results

The results for the weighted Canberra distances for the test set are reported in Table 1, in which the results are tabulated as illuminant, camera, method type, and model. A summary of the results is reported in Table 2, where the metrics are averaged over all cameras to provide an overall perspective. Finally, a random selection of the spectra estimated using the non-negative deconvolution model is shown in Figure 8 for a visual comparison of the estimates in comparison to the ground truth spectra. The results clearly indicated that the deconvolution method performed significantly better than the cepstrum methods. All models performed significantly worse for the cold white LED illuminant, which was not present in the training data, indicating that generalization to illuminants not present in the training set was much weaker. The differences between the camera models were fairly small. For better-performing deconvolution models, the smartphone camera, Samsung S10, performed the best, with a mean weighted Canberra distance of 0.0356 for the non-negative model under halogen illumination, although performance differences between the cameras were negligible. For comparison, the mean weighted Canberra distance was approximately 0.056 between the ground truth spectra and ground truth spectra added with zero mean normally distributed noise with a standard deviation of 10% of the mean of each spectrum.

The plots in Figure 9 show the error dependence of the test set spectra with respect to the wavelength. The logarithmic mean error is calculated for each wavelength λ by:$$\log\left(\frac{\sum \frac{|y\left(\lambda \right)-\hat{y}\left(\lambda \right)|}{y(\lambda )}}{n}\right)$$
where y(λ) is the ground truth spectrum at λ, $\hat{y}\left(\lambda \right)$ the estimated spectrum at λ, and *n* the number of test spectra, i.e., n=36. The maximum is similarly calculated for each wavelength λ by:$$\log\left(\max_{(y,\hat{y})\in \mathrm{Test}\;\mathrm{set}}\left\{\frac{|y\left(\lambda \right)-\hat{y}\left(\lambda \right)|}{y(\lambda )}\right\}\right)$$

For the cepstrum models, the errors were more concentrated than for the deconvolution models, but at a higher level. For the deconvolution models, the mean absolute error was relatively stable across the entire spectrum, with slightly better performance in the 400–450 nm range. The error of the cepstrum models, although on average higher, was relatively more stable over the spectrum than the error of deconvolution models. Within the range of 550–600 nm, in which most cameras had high spectral sensitivity, the cepstrum models performed slightly better. On average, the 0.95 quantile error was approximately 6.25-times higher for the deconvolution models and approximately 4.8-times higher for the cepstrum models than the mean error, indicating a relatively large worst-case deviation from mean performance for all models and methods.

To complement the direct evaluation of spectral reconstruction, we also report accuracy in terms of color specification. Table 3 shows the median color difference values $\Delta E_{00}$ in the CIEL*a*b* space [50] for the different model, camera, and illuminant combinations, and Table 4 summarizes these by each model and illuminant. The metric was designed so that a value of one indicates a color difference that is just perceivable for a typical human observer. We report the medians over the test spectra as a more reliable overall estimate, to avoid giving excess weight for a small number of outlier samples. For $\Delta E_{00}$, the CIEL*a*b* values were calculated from the CIE XYZ tristimulus values, which were calculated from the spectra using the CIE 1931 2^°^ standard observer color matching functions without an illuminant reference because the spectra already included the effect of the illuminant. The $\Delta E_{00}$ values indicated that the cepstrum method was poor at estimating the CIEL*a*b* values, while the deconvolution methods performed significantly better, especially under halogen illumination.

## 7. Discussion and Conclusions

The results indicated that we can extract the spectra of color patches in different illumination conditions using diffraction imaging and algorithmic reconstruction of the spectrum. The estimates were not perfect, and the accuracy depended on the lighting and camera model; however, the important observation was that the general-purpose approach only requiring an optically simple add-on device attached to a regular camera sensor worked for a wide range of conditions with satisfactory accuracy. The method based on the deconvolution operation resulted in more accurate spectrum extraction, compared to the method based on the cepstrum operation, but the cepstrum method may still have value for calibration purposes. Whilst the weighted Canberra distance results for the cepstrum models were somewhat poor, the cepstrum operation itself was merited with its simplicity and partial robustness, as discussed in Section 3.2. Indeed, the original idea for using the cepstrum came from the work in [17] and from diffraction imaging experiments conducted by the authors of metal halide and fluorescent illuminants, characterized by strong spectral peaks, which were immediately evident in the extracted cepstrum profiles. Notable in these experiments was the good agreement of peak location and relative intensity of the cepstrum profile with spectra measured using a calibrated spectrophotometer, after having accounted for the camera’s spectrum response.

The test set data were collected independently of the training data and consisted of paint samples from a different manufacturer, and we used also an illuminant that was not used in the training data. In other words, the testing procedure detailed in Section 5.5 was specifically designed to measure the out-of-domain generalizability of the methods. The results for color patches under halogen illumination indicated how well the models generalized to unseen color patches under an illuminant included in the training set, yet whose spectrum was unknown. The results for color patches under the cold white LED illumination indicated how well the models generalized to unseen color patches under an illuminant not included in the training set, thus representing the more challenging case for out-of-domain generalizability (an even more challenging case would be to adapt a trained model to a new camera from a small number of training samples and test generalizability to unseen color patches and illuminants). For both of these test cases, the deconvolution models provided good results, especially when considering that the ground truth spectra are unlikely to be perfect. Due to the good performance of the deconvolution models for illuminants also found in the training set, we suspect that a pretrained model could be easily adapted for better accuracy for new illuminants with a relatively small number of samples.

When inspecting the results from the perspective of determining the color, the smallest error was obtained with the non-negative deconvolution model using the Samsung S10 camera under halogen illumination, reaching $\Delta E_{00}=1.63$. This almost matched the just-noticeable difference of $\Delta E_{00}=1.0$ and is hence likely to be accurate enough for typical applications in need of low-cost imaging devices, but for, e.g., scientific imaging, the results may not be accurate enough. The performance of the other cameras using the deconvolution models, as seen in Table 3, was also fairly good under halogen illumination. A closer inspection of the estimated spectra (Figure 8) and errors (Figure 9) revealed that the errors were larger towards the 700nm end of the spectrum. This was likely due to the vanishing spectral sensitivity of the cameras toward the ends of the visible spectrum. One potential strategy for improving the accuracy would be to remove the infrared filter present in all cameras used in our experiments to increase the spectral sensitivity for larger wavelengths, but investigating the practical value of this would require replicating all of the data collection and training of the models.

The mathematical model used here assumed noise-free images and could be extended to explicitly model electronic and shot noise produced by the imaging device, possibly in a similar fashion as [9]. The computational models themselves introduced some form of noise resilience due to the redundancy of the large spatial area of the diffraction pattern, so that individual noisy pixels did not directly influence the result, but explicitly modeling the noise could still improve the accuracy especially for the cepstrum method.

Better modeling of the diffraction phenomenon could also improve the accuracy, for instance by relaxing the linearity and isoplane requirements imposed on the diffraction kernel $h_{\lambda }$, combined with a more flexible deconvolution model. Furthermore, better modeling of spatial variability in the illumination intensity in combination with the imaging aperture size, exposure time, and signal amplification could help. A potential improvement could also be found in imposing stronger priors on the spectra, similar to Cheung et al. [7], Parkkinen et al. [11], although the lack of explicit requirements can also be considered an advantage of the proposed method. The priors could also include information on the space of trichromatic camera spectral sensitivity functions [31]. Furthermore, improvements can be made by illuminant estimation [51,52] from the diffraction images, which would allow for direct reflectance spectra estimation without illuminant spectra measurement.

## Figures and Tables

**Figure 1 jimaging-07-00166-f001:**
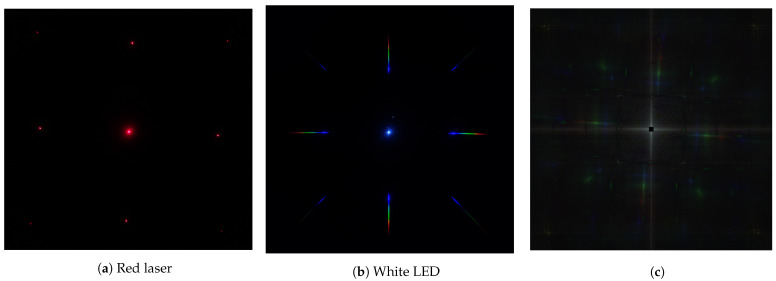
Example diffraction images of point-like light source for (**a**) a red laser and (**b**) a white LED, and (**c**) an example cepstrum image. The diffraction pattern becomes evident when a point-like light source is imaged through a diffraction grating in a dark environment. For (**a**) the red laser, the light is essentially monochromatic, so the diffraction pattern is also point-like. For (**b**), the intensity of the diffraction in for each wavelength is dependent on the diffraction grating efficiency and the spectrum of the light. The diffraction grating used to capture these images is composed of two regular onedimensional diffraction gratings with their axes at approximately a 90° angle. Due to imperfect alignment, the diffraction pattern is not entirely symmetrical. (**c**) The example mean cepstrum image formed from the mean of 60 cepstrum images of a diffused RGB illuminant captured using a Canon M3 camera combined with a diffraction grating, a Sigma 35 mm lens, and a field stop construct attached in front of the lens. The resulting cepstrum image was shifted such that the center of the cepstrum image is displayed in the center of the image. The center of the image was masked out as it contains high intensity pixels that are not relevant to spectrum extraction. Figure best viewed in color.

**Figure 2 jimaging-07-00166-f002:**
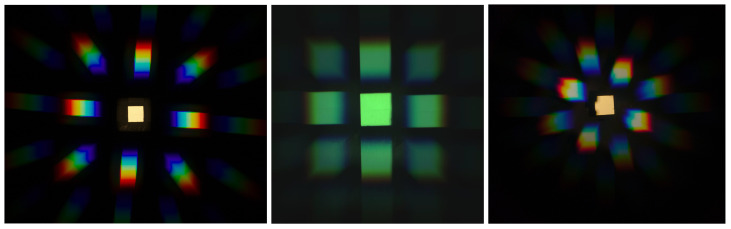
Diffraction images captured using three cameras: Canon M3, Raspberry Pi HQ camera, Samsung S10 main camera. The Raspberry Pi HQ image appears excessively green because it has not been corrected for white balance, which is mostly relevant for presentation purposes only.

**Figure 3 jimaging-07-00166-f003:**
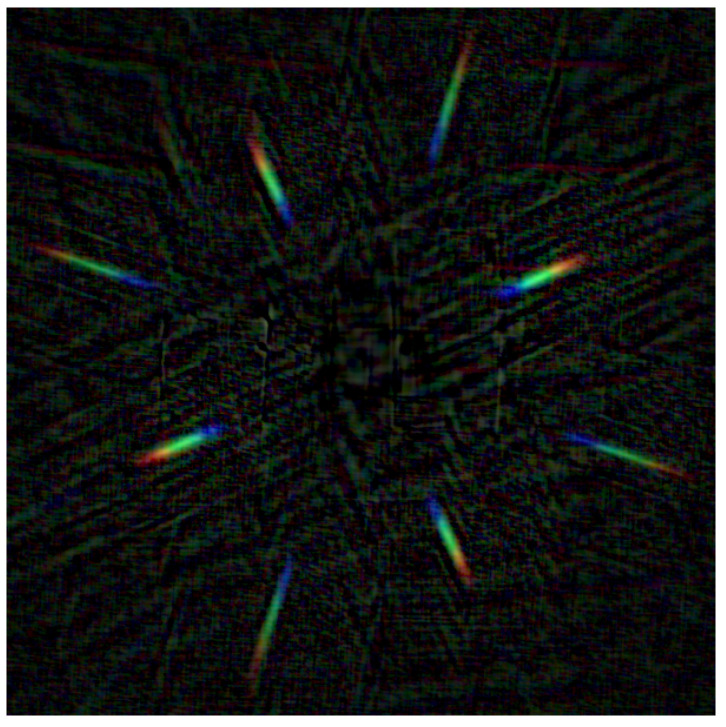
The 2D deconvolution result obtained from a diffraction image using a Dirac delta distribution as the regularization kernel *k*. In this case, the regularization term $|\hat{k}{\left(\xi \right)|}^{2}$ in (Equation 4) is a constant (independent of ξ). The computations were discretized using the discrete Fourier transform.

**Figure 4 jimaging-07-00166-f004:**
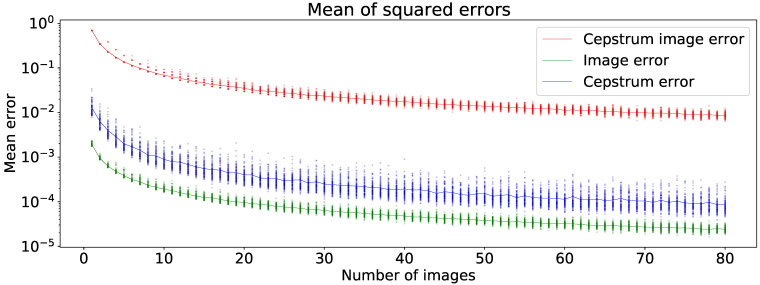
Cepstrum profile and cepstrum image mean-squared errors as a function of the number of cepstrum images from which the mean cepstrum image is computed. The error is the difference between the mean cepstrum image compared to the the mean cepstrum image computed from 140 cepstrum images, i.e., the proxy ground truth. The cepstrum profile error (labeled cepstrum error) is significantly smaller than the cepstrum image error.

**Figure 5 jimaging-07-00166-f005:**
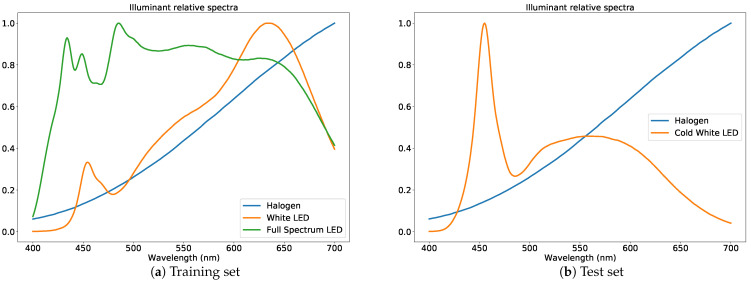
Relative spectra of the illuminants used in capturing diffraction images for the training and test set, respectively.

**Figure 6 jimaging-07-00166-f006:**
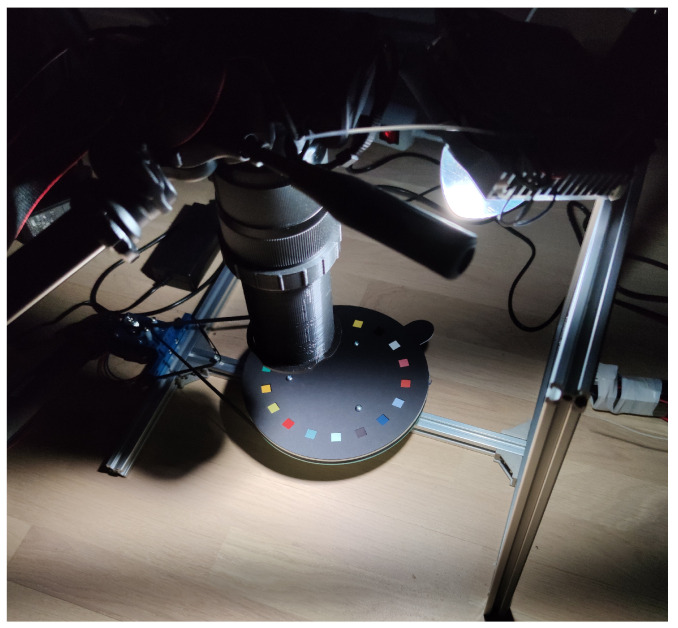
Imaging setup for the Canon M3 camera and the full-spectrum LED illuminant. Other cameras were setup in a similar way, adapting to the camera. The color targets were on a paper disc with a dark mask over each color patch exposing a 1cm by 1cm area of the color patch. The light was mounted to the side of the camera at an angle to reduce the effect of reflection from the surfaces of the color patches. The paper discs were mounted on to a acrylic disc and secured by three screws. The acrylic disc was rotated by a stepper motor via a timing belt.

**Figure 7 jimaging-07-00166-f007:**
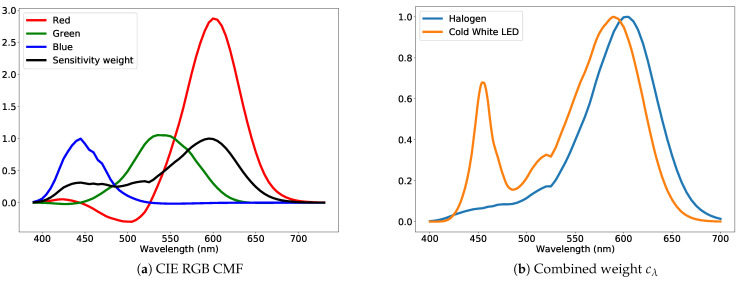
(**a**) CIE RGB color matching functions used as a proxy for camera sensor spectral sensitivity. The mean of the absolute values (in black) was used as a single proxy for sensor sensitivity. (**b**) Combined weight cl calculated from the sensitivity weight and illuminant spectrum.

**Figure 8 jimaging-07-00166-f008:**
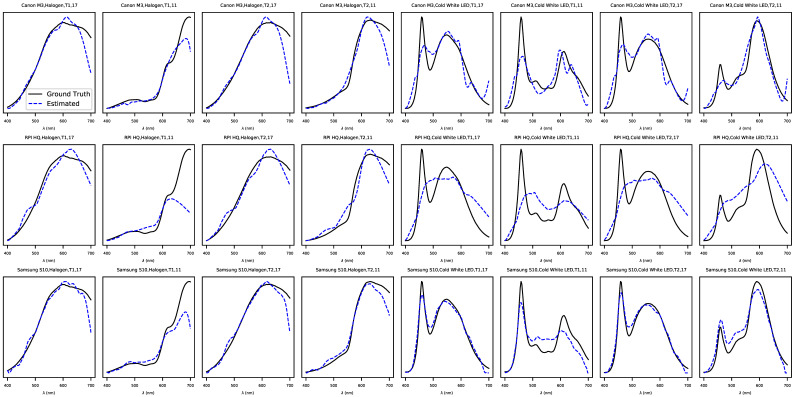
A random selection of relative spectra estimated using the best presented method, the non-negative deconvolution model (blue dashed line) compared against the relative ground truth spectra (black solid line). Each row shows spectra for a single camera with the first half of columns imaged under halogen illumination and the second half under cold white LED illumination.

**Figure 9 jimaging-07-00166-f009:**
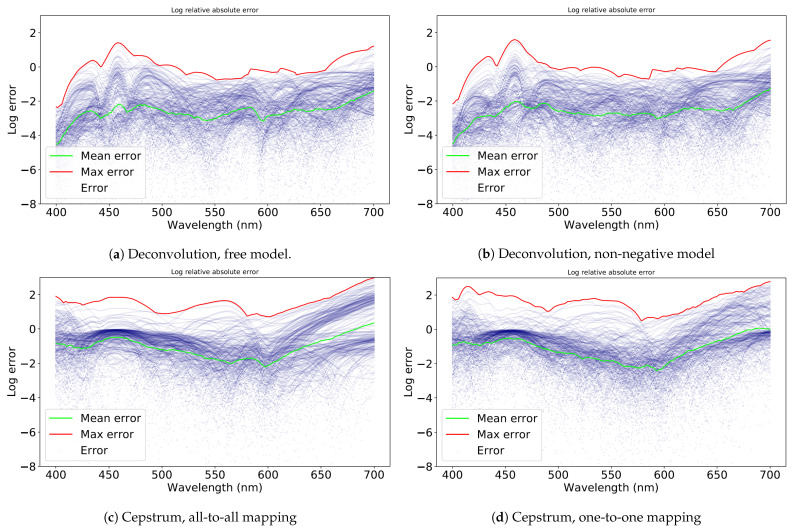
Log relative absolute error of the test set spectra in relation to ground truth spectra. While the mean error for each wavelength is generally low, especially for the deconvolution models, the maximum error can be quite high.

**Table 1 jimaging-07-00166-t001:** Mean of weighted Canberra distances between estimated and ground truth spectra for camera, illuminant, method, and model combinations. Smaller numbers mean better accuracy w.r.t. ground truth spectra. The best result is indicated in boldface.

Mean of Weighted Canberra Distances							
**Illuminant**		**Halogen**			**Cold White LED**		
	Model	Canon M3	RPI HQ	Samsung S10	Canon M3	RPI HQ	Samsung S10
Deconvolution	Free	0.0603	0.0686	0.0359	0.1316	0.1134	0.0586
	Non-negative	0.0552	0.0697	**0.0356**	0.1274	0.1227	0.0592
Cepstrum	Fully connected	0.1089	0.1271	0.1364	0.2663	0.3080	0.2045
	Simple	0.1041	0.1240	0.1272	0.2392	0.2637	0.1995

**Table 2 jimaging-07-00166-t002:** Summary results of the means of weighted Canberra distances between estimated and ground truth spectra for illuminant, method, and model combinations averaged over cameras. The best result is indicated in boldface.

Summary Results: Mean of Weighted Canberra Distances				
**Illuminant**		**Halogen**	**Cold White LED**	
	Model			Mean
Deconvolution	Free	0.0549	0.1010	**0.0779**
	Non-negative	**0.0535**	0.1029	0.0781
Cepstrum	Fully connected	0.1241	0.2591	0.1913
	Simple	0.1184	0.2339	0.1759

**Table 3 jimaging-07-00166-t003:** Median of $\Delta E_{00}$ between estimated and ground truth spectra for camera, illuminant, method, and model combinations. Smaller numbers mean better accuracy w.r.t. ground truth spectra. The best result is indicated in boldface.

Median of $\Delta E_{00}$							
**Illuminant**		**Halogen**			**Cold White LED**		
	Model	Canon M3	RPI HQ	Samsung S10	Canon M3	RPI HQ	Samsung S10
Deconvolution	Free	2.34	3.65	1.70	5.46	4.24	3.55
	Non-negative	2.40	6.94	**1.63**	6.20	10.99	3.38
Cepstrum	Fully connected	12.79	8.57	14.40	28.28	29.65	23.60
	Simple	9.94	11.59	13.05	23.80	26.03	21.97

**Table 4 jimaging-07-00166-t004:** Summary results of the median of $\Delta E_{00}$ between estimated and ground truth spectra for illuminant, method, and model combinations, median over cameras.

Summary Results: Median of $\Delta E_{00}$				
**Illuminant**		**Halogen**	**Cold White LED**	
	Model			Median
Deconvolution	Free	2.44	4.37	3.25
	Non-negative	2.72	6.85	5.08
Cepstrum	Fully connected	12.56	28.56	19.61
	Simple	11.37	23.81	17.49

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
