# Peer review of "Visible Light Spectrum Extraction from Diffraction Images by Deconvolution and the Cepstrum"

_2313-433X, 2021, doi:10.3390/jimaging7090166_

Round 1
Reviewer 1 Report
The paper is well-structured and mathematically sound. Some minor improvements are required:
1) Explain what is ground truth spectra in exact mathematical terms.
2) Add a description of the alternative measures, besides the chosen one (Canberra) and add them for comparison to your experiment. Note that the word "Canberra" is not starting with a capital letter (see line 411 and others). Figure 8 is not explained, as the formulae for Mean, Max error must be provided for self-efficiency of the paper.
3) Formula on page 13 for Y normalized is not well explained.
4) Conclusion is not well-structured or comprehensive.
5) the paper is not well proof-read. See, for example, the sentence on line 480: "The test set data was here collected at a different time to the training set..."
4) Lines 317 through 327 are an interesting material and must be explained earlier and in more detail.
Author Response
1) Explain what is ground truth spectra in exact mathematical terms.
We now provide a full mathematical explanation for the process of forming the ground truth spectra from the reference hyperspectral images in Section 4.3 (Data).
2) Add a description of the alternative measures, besides the chosen one (Canberra), and add them for comparison to your experiment. Note that the word "Canberra" is not starting with a capital letter (see line 411 and others). Figure 8 is not explained, as the formulae for Mean, Max error must be provided for self-efficiency of the paper.
We now define the alternative measures of mean and maximum deviation in Section 6 (Results) and have fixed the capitalization. In addition, we now also define a new metric of \Delta E_{00} in Section 6 (Results), introduced to more directly measure the accuracy of color determination (see the response for Reviewer 2). We note that the exact definition for this metric is not provided in the paper because it is too long and complex. We explicitly refer to Sharma et al. (2005) that provides the detailed definition of this standard metric.
3) Formula on page 13 for Y normalized is not well explained.
We revised the explanation and also now provide full motivation for the normalization.
4) Conclusion is not well-structured or comprehensive.
We have now elaborated more on the shortcomings of our method and possible future improvements to our method in Section 7 (Discussion and conclusion), expanding the section by five full paragraphs. In particular, we now clearly indicate that the idealistic noise-free model of diffraction images is a simplification, and additionally outline possible improvements related to stronger priors for reflectance spectra and camera spectral sensitivity functions.
5) the paper is not well proof-read. See, for example, the sentence on line 480: "The test set data was here collected at a different time to the training set..."
We have carefully proof-read the document for grammar and spelling, making several corrections. We hope the language is now sufficiently clear.
4) Lines 317 through 327 are an interesting material and must be explained earlier and in more detail.
We now discuss the smoothness of reflectance spectra in more detail in Section 4.3 (Data). In addition, we also address smoothness and the PCA encoding of reflectance spectra already in Section 1, in a paragraph that covers related work that explicitly assumes smoothness as part of their solution.
Reviewer 2 Report
The manuscript explores how to get spectral information from diffraction images using a digital camera coupled to a diffraction grating. Two recovery approached are applied: a deconvolution process and a Cepstrum analysis (although it seems that this one was already introduced by the authors in a previous publication). The device would be considered a cheap alternative to other conventional spectroradiometers and spectrophotometers. The accuracy of the measurements are analyzed in terms of the Canberra distance and produces relatively stable errors across the spectrum and good recovery results. Nevertheless the constraints imposed to the main Equation (2) could limit the potential use of the device with more complex objects.
Major comments:
First, the authors claim (on page 1, Abstract) that “no past work…” has investigated the main topic of the work, which is not true. Potential readers can find also early studies already addressing this issue. Here you are a non-exhaustive list of related references, which include examples of different spectral recovery methods, colorimetric and spectral metrics to quantify the algorithm performance, etc.:
- Francisco H. Imai et al., “A Comparative Analysis of -Spectral Reflectance Estimated in Various Spaces Using a Trichromatic Camera System”, Journal of Imaging Science and Technology, 44 (4), 280-377 (2000).
- Vien Cheung et al., “Characterization of trichromatic color cameras by using a new multispectral imaging technique”, Journal of the Optical Society of America A, vol. 22, 7, 1231-1240 (2005).
- Heikkinen et al., “Evaluation and unification of some methods for estimating reflectance spectra from RGB images,” J. Opt. Soc. Amer. A, vol. 25, no. 10, pp. 2444–2458, 2008.
- Peyvandi et al., “Generalized Inverse-Approach Model for Spectral-Signal Recovery” , IEEE TRANSACTIONS ON IMAGE PROCESSING, VOL. 22, NO. 2, 501-510, 2013.
- Shimano and M. Hironaga, “Recovery of spectral reflectances of imaged objects by the use of features of spectral reflectances,” J. Opt.Soc. Amer. A, vol. 27, no. 2, pp. 251–258, 2010.
- Eckhard, T. Eckhard, E.M. Valero, J.L. Nieves, E. Garrote,” Outdoor scene reflectance measurements using a Bragg-grating-based hyperspectral imager”, Applied Optics, vol. 54, N. 13, pp. D15-D24 (2015).
Next, although authors introduced the device as a simple alternative to other expensive spectral devices, no examples are given about a spectral function obtained with the system in comparison with a reference spectrum. The authors state that “Accurate color determination in variable lighting conditions” is achieved. However no color differences (e.g. CIEL*a*b*) are included in the analysis to check the performance of the spectral recovery results.
Other comments:
-On page 1, 1st paragraph, line 18: authors claim that there are no universal mapping from perceived color to other color spaces. Nevertheless quasi-uniform spaces like the CIELAB try to solve for that issue.
-p.1, lines 28-30: No comments about spectral imaging devices are included, such as an RGB camera coupled to a set of color filters, a monochrome camera coupled to different interference filters, etc. (see above recommended references about this issue).
-p.2, l.35: the sentence claiming that your "...approach is the first that does not require special electronic or optical hardware..." needs clarification because it is not clear that your approach could be the first one (see, for instance, Eckhard et al, (2015) where a Bragg-grating-based hyperspectral imager is introduced).
-p.2, l.63: the statement that the system can produce good results “in most conditions” is ambiguous. What does most conditions mean?
-p.3, l.106: Figure 4 is not adequately placed. Revise the order of every figure in the manuscript.
-p.3, Eq.(1): it should be mentioned that illumination is considered totally uniform (i.e. flat). In addition, Eq(1) does not include any noise term. I would like to see some discussion about the effect of noise (white noise, electronic noise, etc.) in that equation and in the following ones.
-p.4, Eq.(2): it is not mentioned that diffraction kernel h_lambda is being considered linear or isoaplanatic throughout the manuscript.
-p.5, 2nd para., l.137: why does “suitable smoothness assumptions” hold?
-p.10, l.244: spectral profiles of the training and test samples should be given.
-p.12, section 4.3., authors state that all possible noise reduction options were disabled but fundamental Eq.(2) does not include any noise influence (see my comment before about this issue). So I wonder if results would have depended on that assumption.
-p.13, 1st-2nd paragraphs: authors should include real examples of some spectral functions recovered by the system in comparison with its reference counterpart. In addition to the Canberra distance, color differences between the test and reference spectral samples should be included to clarify if recovery results are colorimetrically accurate.
-p.13, 3rd para.: authors test the spectra smoothness constraint by applying a PCA to the spectral set and showing that 13 to 23 principal components suffice to represent any spectral function. However, I would expect the opposite, i.e. if only 3 to 8 principal components would suffice to represent the spectral set this means that spectra are relatively smooth.
-p. 17, l.463: what does a value of 0.0356 mean as a good recovery result? Spectral recoveries should be included to illustrate the meaning of that number.
-p. 18, l.460: the statement that “the mean absolute error is relatively stable across the entire spectrum” should be clarified. According to Fig. 8, all errors (including the max errors) seem to change across the spectrum. So I cannot fully understand that relatively stability mentioned by the authors.
-p. 19, Section 7: considering the limitations and constraints imposed to the fundamental Eq.(2) I would like to see some discussion about this issue, how to improved those limitations and the comparison with other potential spectral recovery algorithms.
Author Response
Major comments:
First, the authors claim (on page 1, Abstract) that “no past work…” has investigated the main topic of the work, which is not true. Potential readers can find also early studies already addressing this issue. Here you are a non-exhaustive list of related references, which include examples of different spectral recovery methods, colorimetric and spectral metrics to quantify the algorithm performance, etc.:
Francisco H. Imai et al., “A Comparative Analysis of -Spectral Reflectance Estimated in Various Spaces Using a Trichromatic Camera System”, Journal of Imaging Science and Technology, 44 (4), 280-377 (2000).
Vien Cheung et al., “Characterization of trichromatic color cameras by using a new multispectral imaging technique”, Journal of the Optical Society of America A, vol. 22, 7, 1231-1240 (2005).
Heikkinen et al., “Evaluation and unification of some methods for estimating reflectance spectra from RGB images,” J. Opt. Soc. Amer. A, vol. 25, no. 10, pp. 2444–2458, 2008.
Peyvandi et al., “Generalized Inverse-Approach Model for Spectral-Signal Recovery” , IEEE TRANSACTIONS ON IMAGE PROCESSING, VOL. 22, NO. 2, 501-510, 2013.
Shimano and M. Hironaga, “Recovery of spectral reflectances of imaged objects by the use of features of spectral reflectances,” J. Opt.Soc. Amer. A, vol. 27, no. 2, pp. 251–258, 2010.
Eckhard, T. Eckhard, E.M. Valero, J.L. Nieves, E. Garrote,” Outdoor scene reflectance measurements using a Bragg-grating-based hyperspectral imager”, Applied Optics, vol. 54, N. 13, pp. D15-D24 (2015).
We did not intend to suggest the concept of determining color using light-weight optical elements would be completely new, and all comments relating to lack or previous work referred to the specific technical solution. We do agree, however, that more extensive coverage of different technical solutions for the same problem would be in order. We now extended the discussion on related work substantially in Section 1 (Introduction), explaining several previous methods using e.g. color filters to solve the task (citing also some of the works listed above). To clarify the novelty, we now more clearly explain how we are the first to consider the combination of passive diffraction grating attached to low-cost consumer camera sensors and machine learning methods for estimating the radiance spectrum. This offers notable practical advantages e.g. in terms of fast imaging using short exposure times.
Next, although authors introduced the device as a simple alternative to other expensive spectral devices, no examples are given about a spectral function obtained with the system in comparison with a reference spectrum. The authors state that “Accurate color determination in variable lighting conditions” is achieved. However no color differences (e.g. CIEL*a*b*) are included in the analysis to check the performance of the spectral recovery results.
We focused on evaluating the method in terms of estimating the spectrum itself, rather than color, since we feel it provides a more comprehensive perspective. In the revised version we also evaluate the accuracy of the proposed methods in terms of color specification to provide more direct support for the claim, using the broadly accepted \Delta E_{00} metric in CIELAB color space. These new results are reported in Tables 2 and 4.
Other comments:
-On page 1, 1st paragraph, line 18: authors claim that there are no universal mapping from perceived color to other color spaces. Nevertheless quasi-uniform spaces like the CIELAB try to solve for that issue.
We now mention the CIELAB color space as a good approximation for a perceptually uniform space. While a low-dimensional color space is indeed highly useful for describing color in most cases, we wanted to emphasize that it is still a simplification compared to the full radiance or reflectance spectrum.
-p.1, lines 28-30: No comments about spectral imaging devices are included, such as an RGB camera coupled to a set of color filters, a monochrome camera coupled to different interference filters, etc. (see above recommended references about this issue)
We have extended the discussion on previous methods for color-determination using trichromatic cameras in Section 1 (Introduction).
-p.2, l.35: the sentence claiming that your "...approach is the first that does not require special electronic or optical hardware..." needs clarification because it is not clear that your approach could be the first one (see, for instance, Eckhard et al, (2015) where a Bragg-grating-based hyperspectral imager is introduced).
We have revised the statement that indeed was too broad since “special electronic or optical hardware” is so vague. We intended to emphasize that, to our knowledge, a simple passive combination of a diffraction grating, field stop, and a digital camera has not been previously used for radiance spectra extraction. To our knowledge, the Bragg-grating-based hyperspectral imager introduced in Eckhard et al. (2015) is not a passive device attached to a camera, but rather an integrated hyperspectral imaging system with active components.
-p.2, l.63: the statement that the system can produce good results “in most conditions” is ambiguous. What does most conditions mean?
We changed the phrase to “in some lighting conditions”, since we only empirically evaluate the method in three conditions. We do believe that the method generalizes well since we evaluated it in a setup where no training data for a new lighting condition was available, but more extensive experimentation under several lighting conditions would be needed to justify the original claim.
-p.3, l.106: Figure 4 is not adequately placed. Revise the order of every figure in the manuscript.
We revised the order of figures so that they appear in the same order as the first reference in the manuscript.
-p.3, Eq.(1): it should be mentioned that illumination is considered totally uniform (i.e. flat). In addition, Eq(1) does not include any noise term. I would like to see some discussion about the effect of noise (white noise, electronic noise, etc.) in that equation and in the following ones.
We now explicitly mention in Section 2.2 that we assume the illuminant to be uniform over spatial coordinates, and we also added discussion on the possible sensor noise. We indeed assume in Eq. (1) that the images are noise-free, since the approach seemed to work well already with this simplifying assumption and because the deconvolution method has inherent noise-cancelling ability (see latter comment for details). The revised version mentions electronic and shot noise as examples of sensor noise that could be taken into account in the mathematical model and in Section 7 (Discussion and conclusion) we mention the possibility of extending the work by considering explicit noise models.
-p.4, Eq.(2): it is not mentioned that diffraction kernel h_lambda is being considered linear or isoaplanatic throughout the manuscript.
We now mention in Section 2.2 that the diffraction kernel h_lambda is linear w.r.t. light intensity and that it is isoplanatic and oriented parallel to the imaging sensor.
-p.5, 2nd para., l.137: why does “suitable smoothness assumptions” hold?
The motivation for the smoothness assumption of spectra and camera spectral sensitivities has been added in Section 3.1 (Algorithm 1: Deconvolution). The smoothness assumption for the camera spectral sensitivity is motivated by the Luther condition. The smoothness assumption for spectra is motivated by the assumption that reflectance spectra are often smooth and that the illuminant can be chosen so that it has a smooth spectral power distribution.
-p.10, l.244: spectral profiles of the training and test samples should be given.
We used in total more than 300 different spectra and believe that showing all of them would take up too much space. We did, however, add new Figure 8 that shows 24 examples of randomly selected spectral profiles in the test set. The figure serves two purposes: It shows the ground truth spectra to illustrate what kind of color patches we used, and the estimated spectra (with non-negative deconvolution algorithm) for different camera and illuminant combinations for evaluation of the accuracy of the method.
-p.12, section 4.3., authors state that all possible noise reduction options were disabled but fundamental Eq.(2) does not include any noise influence (see my comment before about this issue). So I wonder if results would have depended on that assumption.
The purpose of disabling possible noise reduction options was to obtain an image signal from the camera sensor with minimal modification and processing by the camera equipment. This way our results would not be influenced by unknown and difficult-to-interpret effects of different noise reduction processes. It is, however, possible that results might have improved if noise reduction options had been enabled.
We also note that the deconvolution method has inherent noise reduction ability, as mentioned in Section 7. The diffraction grating disperses light to a relatively large area (compared to the central image) and the method is computed based on all of the data. The whole diffraction image includes a significant amount of redundancy and we suspect that at least independent pixel-level noise has negligible effect on the end result.
-p.13, 1st-2nd paragraphs: authors should include real examples of some spectral functions recovered by the system in comparison with its reference counterpart. In addition to the Canberra distance, color differences between the test and reference spectral samples should be included to clarify if recovery results are colorimetrically accurate.
Examples of recovered spectra are now shown in Figure 8, which compares them against the corresponding ground truth spectra. We also now report color difference values, as median of \Delta E_{00} in the CIELAB color space, in Tables 2 and 4. We thank you for the valuable suggestion for adding direct comparison also in terms of the color, which indeed makes the results easier to interpret.
-p.13, 3rd para.: authors test the spectra smoothness constraint by applying a PCA to the spectral set and showing that 13 to 23 principal components suffice to represent any spectral function. However, I would expect the opposite, i.e. if only 3 to 8 principal components would suffice to represent the spectral set this means that spectra are relatively smooth.
The number of PCA components required to explain most of the variance in data is not necessarily indicative of the smoothness of the data, because the PCA vectors themselves may not be smooth. We would also like to clarify that the number of PCA components required to explain the variance referred to the results of Jiang et al (2013), not on our training data. For quantifying the smoothness, we used the first and second order differences of the spectra, as explained in Section 4.3.
-p. 17, l.463: what does a value of 0.0356 mean as a good recovery result? Spectral recoveries should be included to illustrate the meaning of that number.
A random sample of recovered spectra is now shown in Figure 8, for visual inspection of the accuracy. Furthermore, for those more familiar with color difference quantification, the \Delta E_{00} values are now shown in Tables 2 and 4. For this new metric the value of one corresponds to just noticeable difference in color, and hence the numbers have a natural interpretation in terms of human perception. While there is no linear mapping between the color difference values and Canberra distances, the two metrics are positively correlated.
-p. 18, l.460: the statement that “the mean absolute error is relatively stable across the entire spectrum” should be clarified. According to Fig. 8, all errors (including the max errors) seem to change across the spectrum. So I cannot fully understand that relatively stability mentioned by the authors.
The relative stability referred to the comparison between the cepstrum and deconvolution models. The deconvolution models show greater performance variability across the spectrum, but also for individual wavelengths. All of the models exhibit considerable performance variability across the range of wavelengths. We now revised the sentence to clarify this.
-p. 19, Section 7: considering the limitations and constraints imposed to the fundamental Eq.(2) I would like to see some discussion about this issue, how to improved those limitations and the comparison with other potential spectral recovery algorithms.
We expanded the discussion in Section 6, by both extending the discussion on the current results and including suggestions for possible improvements to be considered in future research. The accuracy could possibly be improved e.g. by explicit noise models for Eq. (2), more detailed modeling of the diffraction, as well as stronger priors or regularization for spectral smoothness.
Round 2
Reviewer 2 Report
Authors have answered and discussed all my major comments and have included most of them in the fresh version of the manuscript. The color accuracy of the proposed method is still poor in comparison with other methods (particularly in the long wavelength range). Although this issue is discussed on page 21 (lines 533-535) I am not fully convinced about the possible solution proposed in the text. Removing the infrared filter probably will not affect the accuracy in that range and will introduce artifacts in the the color balance of the camera (i.e. you will have to carefully calibrate the camera after that).
Table 2 and 4 captions are ambigous: authors mention that Median of DE00 is shown but in the table (last column) the Mean is mentioned. Do you mean the mean of the median? or the average of DE00?
Author Response
“Authors have answered and discussed all my major comments and have included most of them in the fresh version of the manuscript. The color accuracy of the proposed method is still poor in comparison with other methods (particularly in the long wavelength range). Although this issue is discussed on page 21 (lines 533-535) I am not fully convinced about the possible solution proposed in the text. Removing the infrared filter probably will not affect the accuracy in that range and will introduce artifacts in the the color balance of the camera (i.e. you will have to carefully calibrate the camera after that).”
The color accuracy of the proposed method is indeed not as high as we had hoped for in all of the cases. We went through all of the claims regarding the accuracy and made sure that the claims match the empirical results, to avoid making overly bold claims regarding the accuracy. We now state that regarding color determination the accuracy is likely sufficient for many use-cases one can imagine for low-cost imaging devices, but that the accuracy may not be sufficient for tasks where very high accuracy is needed. We unfortunately cannot directly compare the accuracy against other methods due to fairly large differences in imaging setups and because of many older papers using DE94 and DEab metrics instead of the more modern DE00.
We also clarified the suggestion regarding the removal of the infrared filter. It indeed would also require other changes for calibration, image acquisition, and re-training of all models and hence it remains an open question whether it would help in practice.
“Table 2 and 4 captions are ambigous: authors mention that Median of DE00 is shown but in the table (last column) the Mean is mentioned. Do you mean the mean of the median? or the average of DE00?”
The word ‘Mean’ in the column header of Table 4 was a mistake and we thank the reviewer for pointing this out. All of the reported scores are medians and this has now been corrected.
Finally, we note that we also corrected one mistake not related to these comments. While making a final check we noticed that around lines 523-526 (lines 524-527 in the new version) we accidentally indicated that some color patches were included both in the training and test set. This was not the case, but instead, all color patches in the test set were new.